# GENERALIZABLE 3D EDGE DETECTION FOR SOFT & HARD FEATURES

## ABSTRACT

Understanding 3D objects based on their geometric and physical properties–independent of predefined labels–is essential for creating, modifying, and using the objects in diverse contexts. However, most machine learning approaches in the 3D domain rely heavily on semantic or primitive labeled-data to achieve these tasks. We present a 3D edge detection algorithm that decomposes point clouds into precise geometric components without relying on primitives or semantic labels. This enables us to tackle datasets of freeform, entirely unrestricted objects (as in the Thang3D dataset) that are challenging, and in many cases impossible, for current models in the literature to segment, reconstruct, or produce parametrically. Additionally, we achieve state-of-the-art (SOTA) edge detection accuracy on both the complex Fusion360 Segmentation, Thang3D, and simpler standard ABC benchmarks. Our approach maintains reliable edge detection on soft features where most existing models fail. In addition, when the detected edges are used as input for segmentation, our method outperforms recent segmentation models on intricate geometries. This framework provides a robust and generalizable foundation for edge-aware analysis, segmentation, and generation of diverse 3D shapes well beyond what can be easily labeled by humans.

## 1 INTRODUCTION

Human perception and communication demonstrate how geometric reasoning underlies our ability to interact with the world across modalities. Consider the process of assembling furniture, improvising a bottle opener from a flat-edged tool, or fabricating replacement components with a 3D printer. In vision, diagrams often replace text when furniture assembly defies easy vernacular description. In multimodal contexts such as CAD (computer-aided design) or 3D fabrication, humans construct, segment, and manipulate shapes through purely geometric cues – often initially creating a shape graphically then exporting it to a domain specific language, *not the other way around*. These examples highlight a key principle: humans flexibly decompose objects into meaningful geometric substructures and reason about their functions, often independent of linguistic labels or categorical constraints Shams & Tarr (2002).

In contrast, much of contemporary 3D machine learning (ML)–across reconstruction, generation, and robotic applications–remains constrained by pre-specified shape categories. Such approaches often classify entire objects or their segments into predefined primitives or CAD sequence elements, thereby restricting the scope of representable and analyzable shapes Mo et al. (2019); Engelmann et al. (2020). These constraints propagate through the ML pipeline: reconstruction methods fail when objects contain segments that defy primitive-based categorization; generative models are limited to producing either simple, primitive-composed artifacts suitable for basic 3D printing or complex visual renderings (e.g., Neural Radiance Fields or meshes) tethered to colloquial textual prompts (e.g., "an orange cat"). However, these latter outputs are unsuitable for feature-based manufacturing and offer little support for human editing, as they remain collections of points or unstructured surface elements ("triangle soup") He et al. (2024); Betsas et al. (2025).

Finally, functional understanding under primitive- or label-based paradigms is inherently limited. Shapes are interpreted only through their assigned categories, leaving unlabeled or unconventional geometries without utility. For example, a knife cannot be recognized as a potential bottle opener, despite possessing the requisite sharp edge, and a hook must conform to stereotypical visual expec-

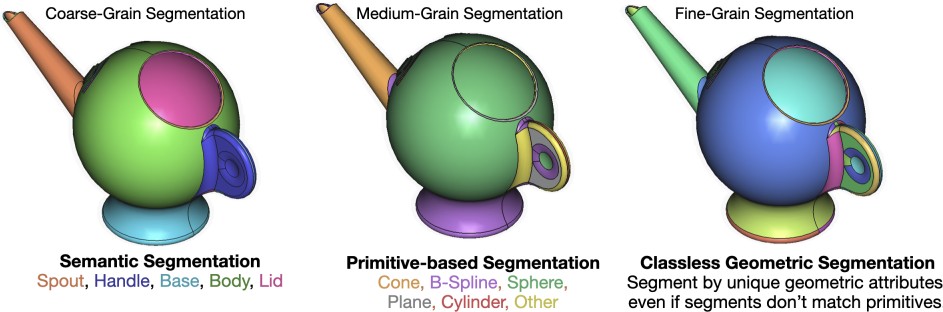

Figure 1: Types of segmentation where (a) shows semantic segmentation of a shape (colors indicate labels), (b) shows class-based geometric segmentation into primitives (colors indicate primitives), and (c) shows classless geometric segmentation (every segment has a unique color).

tations rather than to context-specific geometric affordances. Thus, current ML approaches fall short of the human ability to flexibly reason about geometry and function beyond predefined labels.

Classless geometric segmentation directly addresses these limitations by shifting the focus away from rigid labels and toward the intrinsic structure of shapes, enabling a more flexible and general understanding of geometry that better supports functional, constructive reasoning. This method partitions a 3D model in any representation into meaningful subregions or "parts" using purely geometric and topological cues (curvature, thickness, geodesic distance, skeleton structure, etc.). Unlike semantic (Mo et al. (2019); Engelmann et al. (2020); He et al. (2024); Betsas et al. (2025)) or primitive-based (Nguyen & Le (2013); Shamir (2008); Attene et al. (2018)) approaches that attempt to fit pieces of a shape to known labels or primitive classes, pure geometric segmentation works directly on the surface or volume, grouping portions of the shape into clusters whose boundaries align with salient features such as ridges, valleys, and protrusions. Often performed by humans when creating or recreating a shape, classless geometric segmentation serves as a fundamental pre-processing step in nearly every domain that manipulates or analyzes geometric data – from CAD and graphics to biomedical imaging to scientific visualization – where precisely identifying and understanding each geometric component of the object, rather than the semantic label of each segment, is paramount.

As seen in Figure 1, primitive, class-based segmentation (either semantic or geometric) tends to operate at a much coarser-grained scale where a singular 'segment' in their case can have multiple geometric pieces that should be further broken down to achieve the granularity shown in item (c). These pieces shown in (c) might not necessarily have a logical semantic or categorical label. Additionally, in real world data, these geometric segments can be very small, such as a bevel or fillet comprising of only fraction of the object's total volume or surface area. The accurate capturing of those small segments has historically proven challenging for segmentation models. While it is often visually obvious to a human where a boundary line can be drawn, those deductions must be extracted through pure physical properties of the shape – such as a change in curvature or inflection point – no matter how small the total size of the segment is relative to the original shape Nguyen & Le (2013); Shamir (2008). We introduce a **primitive-less method** of decomposing shapes based purely on geometry, capable of handling **arbitrarily fine features** in **very large point clouds**. This allows us to accurately outline then segment exceptionally diverse shapes.

## 2 RELATED WORK

**Edge Detection** Most edge detection has primarily focused on detecting sharp edges, not smooth transitions. Most real world man-made objects, however, tend to have bevels, chamfers, or fillets (shown in Figure 2) as these provide crucial structural and safety features (truly sharp edges are usually unique to where it necessitates). Meanwhile, sharp edges are practically nonexistent in tasks related to more organic objects, like topology optimization and medical segmentation. In order to accurately segment, reconstruct, or generate these sorts of shapes, smooth edge detection must also be possible. An overview of desirable model capabilities is shown in Table 1.

As the baseline comparison in this paper, **PCEDNet** Himeur et al. (2022) introduces a multi-scale *Scale-Space Matrix* (SSM) descriptor per point (differential shape cues across radii) and a

| Method | Primitive-less | Soft features | Unlimited Points | Arbitrary Resolution | Points Only |
|--------|:---:|:---:|:---:|:---:|:---:|
| DEF [2022] | ✗ | ✗ | ✗ | ✗ | ✗ |
| PIENet [2020] | ✗ | ✗ | ✗ | ✓ | ✓ |
| SEDNet [2023] | ✗ | ✗ | ✗ | ✗ | ✓ |
| PCED [2022] | ✓ | ✓ | ✓ | ✗ | ✗ |
| NerVE [2023] | ✗ | ✗ | ✓ | ✗ | ✓ |
| Ours | ✓ | ✓ | ✓ | ✓ | ✓ |

Table 1: Abilities of edge-detection methods in point clouds, highlighting whether the initial extraction relies on primitives or not, soft feature detection, scalability to unlimited points, handling of arbitrary resolution, and reliance on point-only input (no normals). Our method provides all 5 capabilities, making it highly generalizable.

lightweight MLP/CNN that classifies *edge*, *near-edge*, and *non-edge* points. It precomputes SSM features at 4–128 scales, and the released code reports classification of millions of points in seconds with small training sets. Datasets include their 'Default' shapes, ABC CAD patches, and a SHREC curve benchmark converted to point clouds. Our other baseline, **NerVE** Zhu et al. (2023) learns a *neural volumetric edge* grid whose voxels store occupancy, orientation, and offsets; the grid is converted to a piecewise-linear graph via simple search, then spline-fitted.

**EC-Net** Yu et al. (2018), a precursor to PCEDNet, is a PointNet++ patch-based network that upsamples and consolidates points while being explicitly edge-aware. It regresses both (a) residual coordinates and (b) point-to-edge distances, and uses a joint loss $L = L_{\text{surf}} + \lambda_1 L_{\text{edge}} + \lambda_2 L_{\text{rep}} + \lambda_3 L_{\text{reg}}$, where $L_{\text{rep}}$ encourages even spacing and $L_{\text{reg}}$ is a truncated regression for distances. The model is trained on virtual scans of ShapeNet-like meshes with manually annotated polylines for edges, with patches extracted consistently at train and test time; KNN in $L_{\text{rep}}$ typically uses $K = 4$, and the PointNet++ backbone features $D = 256$ per point.

Other less-generalizable baselines include primitive-based models. **DEF** Matveev et al. (2022) predicts a *distance-to-edge* scalar field on local patches, then fuses patch fields to scale to large clouds; feature curves are extracted by following field minima and fitted as parametric primitives. Supervision is based on distance to the nearest sharp curve, the model is trained on synthetic CAD (ABC) and then fine-tuned on scans, and the outputs feed a parametric curve reconstructor. **PIE-NET** Wang et al. (2020) formulates edges as a set of parametric curves (lines, circles, B-splines). A region-proposal stage over-generates edge and corner candidates, and a ranking stage selects a consistent subset and fits primitives end-to-end. **SEDNet** Li et al. (2023) is a two-stage fusion network that labels surface/edge points to drive geometric primitive fitting (planes, cylinders, etc.) AGPN Ni et al. (2016) detects edges using neighborhood geometry (RANSAC + angular gap) and then traces feature lines via region growing/model fitting–an important non-learned baseline and evaluation reference.

We evaluate against PCEDNet and NerVE as they are both newer improvements on DEF and are capable of processing much larger point clouds than DEF in a practical amount of time. DEF states it only extracts sharp features, and our interest is in extracting very fine details, both sharp and soft. This requires very densely scanned point clouds akin to what would be seen in industrial scanning tools which can create millions of points at a time on a single object Franaszek et al. (2024).

**Segmentation** Two works relevant to recent advances in geometric segmentation are ParSeNet Sharma et al. (2020) and SpelsNet Cherenkova et al. (2024) – which attempt to reconstruct boundary representation (BREP) files from discrete forms. ParSeNet extends the above paradigms by decomposing point clouds into parametric surface patches–including B-spline and primitive models–within an end-to-end trainable framework, improving segmentation fidelity and producing robust parametrizations for shapes with clear primitives. SpelsNet, building on ParSeNet, jointly leverages both spatial and topological cues: a sparse-convolutional encoder feeds into (1) a spatial head that classifies each point's primitive type (e.g. planar face, cylindrical face, line-edge, spline-edge) and learns metric embeddings for grouping points into coherent surface or curve elements, and (2) a graph-based head that, via a novel point-to-BREP adjacency formulation, directly supervises the Linear Algebraic Representation (LAR) of the underlying BREP chain complex. Both these datasets primarily train on the ABC parts dataset, and unfortunately the SpelsNet does not release CC3D-VEF dataset or else it is no longer available online. Both models are still primitive-driven in their segmentation and reconstruction pipelines.

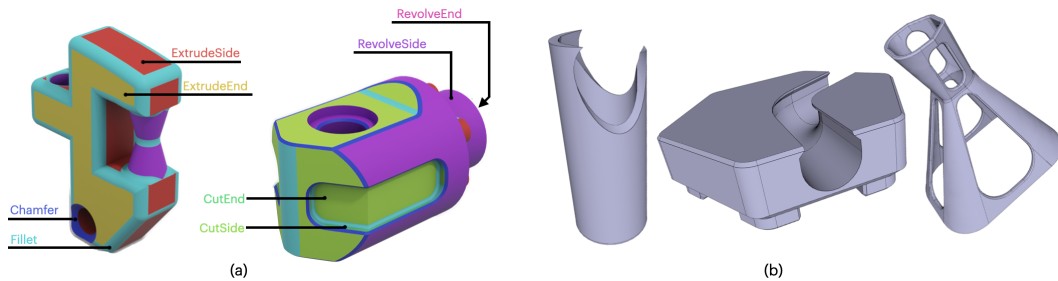

Figure 2: (a) soft features (fillet, chamfer, revolve – features without a sharp change in curvature) and hard features (cut, extrude – features with a sharp change in curvature) from the F360+ dataset Lambourne et al. (2021), (b) complex, unrestricted shapes from the Thang3D dataset.

## 3 EXPERIMENTAL SETUP

We demonstrate high-fidelity, classless edge-detection on variable-sized point clouds (sampled from remeshed versions of BREP files) into their respective fine grain boundary-representation segments (as shown in Figure 1 with their soft-features intact). Proper geometric-based edge detection and segmentation allows subsequent precise reconstruction of diverse shapes (e.g. using the reconstruction pipeline in Point2CAD Liu et al. (2024)) or far more diverse generation when extracted segments are used as input features (e.g. using Xu et al. (2024)). The starting BREP representation provides ground truth for metrics to evaluate edge detection performance, but features are extracted from discrete mesh, point cloud, or neural radiance field representations. Even when starting only with points and then the approximating normals, the use of other geometric features deterministically computed allows us to maintain fairly robust edge-detection.

### 3.1 EVALUATION DATASET

CAD datasets of boundary representation files in either BREP or STEP form are suitable for our task as they provide precise geometric breakdowns well beyond semantic labels; however, most 3D CAD generation datasets for ML (e.g., Wu et al. (2021), Colligan et al. (2022), Xu et al. (2024)) intentionally lack soft complex features that are necessary for our evaluation. We therefore chose the Fusion360 Segmentation BREP dataset since it has soft, complex features as well as extremely fine-grained labeling of geometric segments. Originally presented by Autodesk for segmentation on the BREP files themselves, we use a modified version of this dataset that was presented by Kimmel et al. (2025) called F360+ and the complex Thang3D dataset. Thang3D shapes were created by humans and have no restrictions on what CAD operations, number of segments, or geometric primitives are allowed to exist in the shape.

An important feature of our model is that it takes an arbitrarily large set of points and classifies them in batches, iteratively building a global context vector that informs each next batch of points. This allows us to process much larger point clouds (>200k points) that grow in size as the geometries become more complex. Through curvature pre-processing, we densely oversample high-curvature regions to accurately capture rapidly changing geometry, allowing us to identify finer features like chamfers, fillets, and bevels that might otherwise be missed by smaller, coarser point clouds.

### 3.2 TASK DEFINITION

We perform edge detection on a dense point cloud, where points and normals are denoted as $X, Y, Z, N_x, N_y, N_z$ and can either be sampled from either a 'perfect' surface descriptor such as a boundary representation file or a more noisy approximation such as a mesh or neural radiance field. We generate datasets of surface point clouds and normals sampled from meshes that approximate BREP files. From the normals, a deterministic pre-processing algorithm approximates both the local curvature, denoted as $H$, and the gradient of the curvature, denoted as $\nabla H$, at various sample sizes. These inputs are fed in sets of 10k points iteratively into the model. A custom deterministic clustering algorithm is then applied to group enclosed boundary regions into designated segments. An outline of the overarching process is shown in Figure 3.

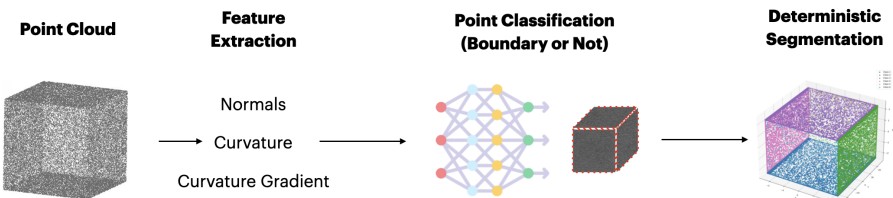

Figure 3: General approach for shape processing: starting from a raw point cloud, features such as normals, curvature, and curvature gradients are extracted, followed by classification into boundary/non-boundary points, which enables deterministic segmentation of the shape.

## 3.3 FEATURE SPACE

Surface points, normals, curvatures and their gradients are input into a convolutional neural network. It is notable that with a minimal amount of deterministic feature extraction as a pre-processing step, one can leverage a far less complex architecture and still achieve state of the art results.

Table 2 summarizes the geometric features used as model inputs. Surface points and their normals define the local representation, while neighborhoods $\mathcal{N}_k(i)$ provide context for computing local properties. Mean curvature is estimated by measuring the divergence of normals, yielding a scalar that encodes how strongly the surface bends at a point. The curvature gradient extends this by quantifying how curvature varies across neighbors: its vector form captures direction and magnitude, while the signed magnitude $s_i$ incorporates whether curvature is increasing or decreasing. Together, these equations provide a compact yet expressive description of local surface geometry.

| Feature | Definition | Equation |
|---|---|---|
| Surface Points | Point cloud with normals | $\{(p_i, n_i)\}_{i=1}^N \subset \mathbb{R}^3 \times S^2,\ \|n_i\| = 1$ |
| Neighborhood | $k$-nearest neighbors of $p_i$ | $\mathcal{N}_k(i) = \{j_1, \ldots, j_k\}$ |
| Mean Curvature | Discrete divergence of normals | $\operatorname{div} n(p_i) \approx \frac{1}{k}\sum_{j \in \mathcal{N}_k(i)} \frac{(n_j - n_i)\cdot(p_j - p_i)}{\|p_j - p_i\|^2 + \varepsilon}$ |
| | Estimated curvature | $H(p_i) \approx \frac{1}{2}\left|\operatorname{div} n(p_i)\right|$ |
| Curvature Gradient | Discrete gradient | $\hat{\nabla} H(p_i) = \frac{1}{k}\sum_{j \in \mathcal{N}_k(i)} \frac{\Delta H_{ij}}{d_{ij}^2}\, \Delta p_{ij}$ |
| | Definitions | $\Delta p_{ij} = p_j - p_i,\quad \Delta H_{ij} = h_j - h_i,\quad d_{ij}^2 = \|\Delta p_{ij}\|^2 + \varepsilon$ |
| | Signed magnitude | $s_i = \operatorname{sign}\!\left(\frac{1}{k}\sum_{j \in \mathcal{N}_k(i)} \Delta H_{ij}\right)\|\hat{\nabla} H(p_i)\|_2$ |

Table 2: Geometric feature definitions and equations, pre-computed per-point for model inputs.

## 4 APPROACH

**Model Architecture** Our model shown in Figure 4 is designed to process very long sequences of 3D points and their normals by splitting them into subset chunks and carrying forward a learned "context" vector that captures global shape information. Each chunk of 10k points that is processed in two parallel branches: a **main branch** that sees all eleven channels where the per-point input is of the form:

$$[X, Y, Z, N_x, N_y, N_z, H_{s10}, H_{s20}, \hat{\nabla} H_{s5}, \hat{\nabla} H_{s10}, \hat{\nabla} H_{s20}]$$

and a **skip branch** that attends only to a subset of five "late-arriving" features that focus on curvature:

$$[H_{s10}, H_{s20}, \hat{\nabla} H_{s5}, \hat{\nabla} H_{s10}, \hat{\nabla} H_{s20}]$$

where $H_{sN}$ indicates curvature approximated by the change in normals across a sample size of $N$ points and $\hat{\nabla} H_{sN}$ approximates the gradient of the curvature across a sample size of $N$. Both branches consist of a 1×1 convolution (effectively a learned per-point linear projection), followed by batch normalization and a ReLU nonlinearity; each produces a 64-dimensional per-point embedding.

These two 64-dimensional embeddings are concatenated to yield a 128-dimensional feature vector at each point, which is immediately "fused" back down to 64 channels via another 1×1 convolution,

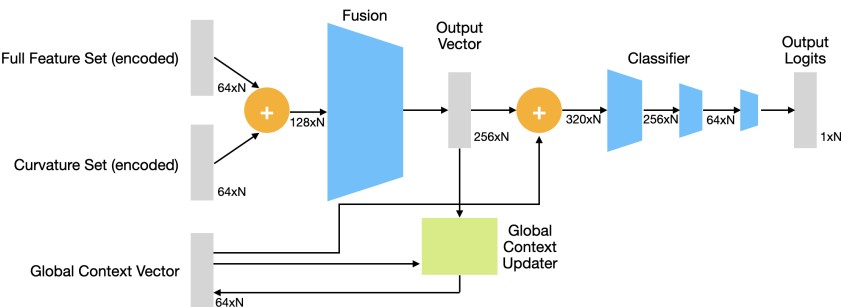

Figure 4: Model architecture where a point cloud of arbitrary size is input in chunks. We combine a main branch (all features) with a skip branch (focusing on curvature) to capture both global and local details. It fuses these representations, encodes them into higher-level features, and integrates a dynamic context vector that updates with each chunk. The final classifier predicts per-point logits.

batch norm, and ReLU. This fused representation is then passed through a small encoder made of two successive blocks of 1×1 conv → batch norm → ReLU, expanding the channel dimension first to 128 and then to 256. In this way, each point's local information is gradually lifted into a richer 256-dimensional space, where higher-order interactions among the original features can be captured.

To bring in information about the shape processed so far, the network maintains a **context vector** of dimension $D$ (default $D = 64$). Before classification, this vector is tiled across all points in the current chunk and concatenated with the 256-dimensional encoded features to yield a $(256 + D)$-dimensional per-point representation. A 1×1 convolutional "context fuser" then projects this back down to 256 channels, normalizes, and applies ReLU, effectively allowing the global summary to modulate each local descriptor. A lightweight classifier head–again two 1×1 convolutions separated by batch norm and ReLU–finally reduces the per-point feature to a single logit, producing a score for each of the up to 10k points in the chunk.

Between chunks, the model updates its context vector so that information can flow across chunk boundaries without requiring the entire sequence to be resident in GPU memory. Concretely, the pooled maximum over the 256-dimensional encoded features (i.e., a per-channel max-pool across all points in the chunk) is passed through a small 1×1 conv → batch norm → ReLU "context update" module to produce a new $D$-dimensional vector. This new context is averaged with the incoming context to form the "updated" context, which is then carried forward to the next chunk.

At inference time, the full point sequence is split into non-overlapping 10k-point chunks; the initial context is set to zero. Each chunk is processed via the above pipeline, yielding per-point logits and an updated context, which seeds the next chunk. Finally, the per-point logits are concatenated in order, yielding a global score for every point in the original sequence, along with the final context vector summarizing the entire shape. This architecture thus balances fine-grained local modeling (via per-point convolutions) with coarse, global information flow (via the recurrently updated context vector), all without any spatial downsampling or dropout, making it well-suited for tasks that demand precise point-level labeling across very long contours.

**Deterministic Segment Clustering: Flood-Fill**   We used a GPU-based algorithm for geodesic-style segmentation of 3D point clouds. The method requires only the raw point coordinates $P \in \mathbb{R}^{N \times 3}$ and per-point normals $N \in \mathbb{R}^{N \times 3}$. Affinities, distances, and segmentation decisions are derived solely from $(P, N)$. No mesh connectivity, surface reconstruction, or external priors are needed. We construct a boundary-aware clustering of point clouds by modifying a KNN graph to respect boundary constraints. Given points $P \in \mathbb{R}^{N \times 3}$, normals $N \in \mathbb{R}^{N \times 3}$, and a boundary indicator $B \in \{0, 1\}^N$, we compute nearest neighbors using a KD-tree and distances to boundaries $d_i = \text{dist}(p_i, \partial)$ via a KD-tree built on boundary points. Candidate edges $(i, j)$ from the $k$-nearest neighbors are pruned if they cross boundary sets ($B_i \oplus B_j = 1$), if $\min(d_i, d_j) \leq \tau_{\text{reject}}$, or if the midpoint $m = \frac{1}{2}(p_i + p_j)$ satisfies $\text{dist}(m, \partial) \leq \tau_{\text{reject}}$.

Surviving edges are assigned a base weight $w = \|p_j - p_i\|_2$. If any of $\{d_i, d_j, \text{dist}(m, \partial)\} \leq r_{\text{bdry}}$, we apply two penalties. First, an angular penalty: if $\theta = \arccos(\text{clip}(n_i^\top n_j, -1, 1)) > \theta_0$, then $w \leftarrow w \cdot \left(1 + \lambda \frac{\theta - \theta_0}{\pi - \theta_0}\right)$. Second, a proximity penalty: letting $d_{\min} = \min(d_i, d_j, \text{dist}(m, \partial))$, we set $w \leftarrow$

$w \cdot \max\left(1, \frac{r_{\text{bdry}}}{\max(10^{-12}, d_{\min})}\right)$. The resulting edges are symmetrized and assembled into an adjacency matrix, either binary (default) or weighted, from which connected components yield cluster labels $\ell \in \{1, \ldots, C\}^N$. The algorithm guarantees that clusters do not cross boundary/non-boundary sets, rejects edges too close to boundaries, and adaptively penalizes edges near boundaries through angular disagreement and inverse-distance factors. In the absence of boundaries, the procedure reduces to standard kNN connected components. The overall complexity is $O(N \log N + Nk)$. The entire algorithm is described holistically in the appendix in Algorithm 1.

## 5 EXPERIMENTS

**Edge Detection** For boundary point classification, we compare our work against NerVE (2023) and PCED (2022) for a subset of 5k parts from ABC, as well as the entire F360+ and Thang3D datasets. We train only on the F360+ dataset, with no fine-tuning for testing on the ABC and Thang3D datasets (see Table 3, containing accuracy, F1, precision, and recall). Output outlines can be seen in Figure 5, with more in the appendix in Figure 7.

For NerVE, the most recent edge detection method, precision, recall, and accuracy are somewhat difficult to compute since the output of NerVE is a parametrized curve. The original paper reported precision and recall for 'edge

| Method | Dataset | ↑ Acc. | ↑ F1 | ↑ P | ↑ R |
|--------|---------|--------|------|-----|-----|
| NerVE | ABC | 0.70 | 0.42 | 0.42 | 0.34 |
| | F360+ | 0.76 | 0.34 | 0.41 | 0.25 |
| | Thang3D | 0.75 | 0.32 | 0.39 | 0.24 |
| PCED | ABC | 0.95 | **0.89** | **0.95** | **0.82** |
| | F360+ | 0.94 | 0.51 | 0.36 | 0.82 |
| | Thang3D | 0.31 | 0.17 | 0.11 | 0.23 |
| Ours | ABC | **0.98** | 0.81 | 0.81 | 0.82 |
| | F360+ | **0.98** | 0.87 | 0.85 | **0.90** |
| | Thang3D | **0.97** | **0.78** | **0.81** | **0.77** |

Table 3: Edge detection performance across datasets: prior methods degrade on more complex data, while ours remains robust.

occupancy' per voxel in their cube grid, that is, a binary classification 'is there an edge in this voxel' which is not an accurate measure of outlines. For consistency with ours and PCED, we sample from their reconstructed parametric curves and compare their sampled points to our sampled points of the original curves. In Table 3 a point is considered 'correct' if it is within 0.05 (i.e. 2.5% of the shape dimensions, since it was scaled to within the unit sphere) of the original points and incorrect otherwise. This is a generous threshold, performance rapidly degrades as we tighten the threshold as shown in the appendix in Table 6. PCED only provides a compiled binary, which we tested on 150 shapes from each non-ABC dataset (ABC values are taken from the original paper). A significant portion of complex shapes resulted in crashing code, possibly since they are too complex. Numbers reported are for shapes that successfully ran. Only 61.3% of F360+ and 24% of Thang3D shapes were fully processed.

While both prior works perform well on relatively simpler shapes that have clean edges, they fail to detect edges that include soft features, including bevels, chamfers, and fillets. They perform exceptionally poorly on 'real life' shapes from the Thang3D dataset.

**Ablation Studies** Ablation studies in Table 4 were performed on the F360+ dataset, highlighting the contribution of different geometric signals to the overall performance. The full model ("Original") achieves the strongest results, maintaining high accuracy (0.98) alongside balanced F1, precision, and recall across categories. Removing curvature gradients results in only a moderate drop, particularly for "many extrudes" where recall declines, but the model still preserves relatively strong overall performance. Architecturally, discarding the global context vector leads to a sharper degradation, especially in recall (0.62 overall), underscoring its importance for capturing broader structural patterns. The most severe impact arises from using only points and normals, where overall F1 falls to about 0.61 and performance on "many extrudes" drops



Figure 5: Thang3D shapes outlines with soft features.

substantially (0.41 F1), suggesting that this representation alone is insufficient despite being exceedingly popular in prior work. Finally, the approximated normals variant (accuracy 0.97, 0.73 F1) performs between the "no curvature gradients" and "only points and normals" settings, indicating

| Ablation | Category | Epochs | ↑ Accuracy | ↑ F1 | ↑ Precision | ↑ Recall |
|---|---|---|---|---|---|---|
| Original | Overall | 28 | 0.98 | 0.87 | 0.85 | 0.90 |
| | Soft features | 28 | 0.98 | 0.89 | 0.87 | 0.91 |
| | Many extrudes | 28 | 0.97 | 0.76 | 0.79 | 0.74 |
| Only Points & Normals | Overall | 11 | 0.95 | 0.61 | 0.60 | 0.66 |
| | Soft features | 11 | 0.96 | 0.63 | 0.66 | 0.64 |
| | Many extrudes | 11 | 0.98 | 0.41 | 0.50 | 0.37 |
| No Curvature Gradients | Overall | 9 | 0.97 | 0.81 | 0.80 | 0.83 |
| | Soft features | 9 | 0.96 | 0.83 | 0.83 | 0.84 |
| | Many extrudes | 9 | 0.97 | 0.70 | 0.76 | 0.66 |
| No Global Context | Overall | 35 | 0.97 | 0.68 | 0.75 | 0.62 |
| | Soft features | 35 | 0.97 | 0.67 | 0.75 | 0.61 |
| | Many extrudes | 35 | 0.95 | 0.60 | 0.67 | 0.54 |
| Approximated Normals | Overall | 83 | 0.97 | 0.73 | 0.72 | 0.75 |
| | Soft features | 83 | 0.97 | 0.70 | 0.71 | 0.68 |
| | Many extrudes | 83 | 0.96 | 0.70 | 0.70 | 0.69 |
| Noise* | Overall | 13 | 0.98 | 0.83 | 0.83 | 0.83 |
| | Soft features | 13 | 0.98 | 0.87 | 0.87 | 0.86 |
| | Many extrudes | 13 | 0.97 | 0.63 | 0.69 | 0.60 |

Table 4: Performance of our model on the F360+ dataset with different ablation settings across categories. Noise values of up to 0.002 (assuming the parts, normalized to the unit sphere, are two inches in maximum dimension) is similar to that of commercial 3D scanners in midrange conditions, where commercial scanning abilities range at the high end from 5-50 microns in error to the hobbyist level 0.1-1mm in error, per Franaszek et al. (2024).

that while normals remain a useful signal even when estimated, their quality critically affects downstream predictions. Together, these comparisons show that curvature gradients and global context substantially boost performance, while high-fidelity normals are essential for robust generalization.

**Segmentation** We compare our segmentation model against that of ParSeNet, which was built on top of PointNet++. It is the one of the current SOTA models for open-source for 3D geometric part segmentation to our knowledge as SpelsNet did not release their code, and most other models (ex: SAMPart3D Yang et al. (2024)) deal in semantic segmentation.

Our final output averages in Table 5 span all shapes (including those without special features) and are averaged over 4 runs. Our method performs significantly better across nearly all categories, with the exception being the number of missing/false segments in ParSeNet's average. This is likely because ParSeNet has a maximum cluster number that never changes and allows small fringe clusters to form, some of which will completely overlap with where a segment should be even if the cluster itself is small and largely incorrect. It also intrinsically limits the number of false segments to the maximum cluster number (50). In contrast, ours has no maximum cluster amount and requires a certain number of cluster points to be considered a valid cluster. Both of the ParSeNet variants average lower intersection over union scores. We see a sub-

| | Category | ↑ mIoU | ↓ M. Seg | ↓ F. Seg10 |
|---|---|---|---|---|
| ParSeNet+ | Fillets | 0.65 | 8.37 | 3.11 |
| | Chamfers | 0.70 | 6.17 | **2.81** |
| | Revolve | 0.65 | 4.09 | **1.69** |
| | > 7 Extr. | 0.54 | 7.75 | 14.54 |
| | Average | 0.64 | 5.86 | 3.48 |
| ParSeNet | Fillets | 0.60 | **0.53** | 7.80 |
| | Chamfers | 0.63 | **0.57** | 7.03 |
| | Revolve | 0.64 | 0.83 | 3.31 |
| | > 7 Extr. | 0.52 | **0.09** | 25.54 |
| | Average | 0.60 | 1.13 | 5.21 |
| Ours | Fillets | **0.76** | 1.21 | **2.27** |
| | Chamfers | **0.77** | 1.42 | 3.41 |
| | Revolve | **0.69** | 0.73 | 3.10 |
| | > 7 Extr. | **0.87** | 2.64 | **9.64** |
| | Average | **0.83** | 0.70 | **3.32** |

Table 5: Segmentation Metrics: mIoU = matched mean IoU, M. Seg = # of missing segments, and F. Seg10 = # of false segments, counting matches with ≥10% overlap.

stantial improvement in recovering true segments for shapes with greater than 7 extrusions, likely due to the fact we can segment a shape with an arbitrary number of segments while again ParSeNet has a limited number of clusters that can form. Results also show that the ParSeNet variants perform significantly worse in regards to missing segments with normal vectors (ParSeNet+) than without normal vectors (ParSeNet) on the points, indicating that at the very least the processing of the nor-

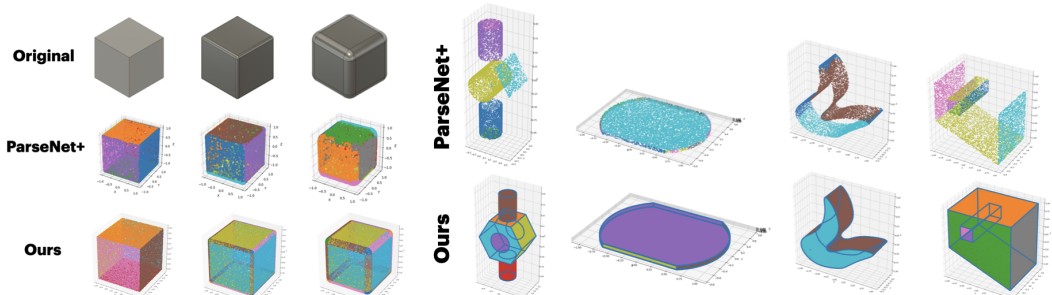

Figure 6: Improvements using our segmentation method in capturing both fine, soft features like fillets (left) as well as simpler features and free-form splines (right) compared to ParSeNet+.

mal feature is severely overfit to simplified data. Additionally, our method is able to recover the true boundaries of regions with a much higher fidelity than ParSeNet, denoted in blue in Figure 6 on the right. Recovering the true boundary is important in any application that requires parametric surface reconstruction, such as CAD design or medical spline-fitting. Our method performs better both on 'soft features' with free-form curves and fillets as well as hard prismatics – including many extrusions.

## 6 CONCLUSION

We introduced a classless 3D edge-detection framework that directly decomposes point clouds into fine-grained geometric components without reliance on labels or primitives. Our method leverages curvature and curvature-gradient inputs, combined with a global context vector that scales to large point sets, enabling high-fidelity segmentation of both hard and soft features. In doing so, we address key limitations of existing state-of-the-art approaches–namely, their inability to robustly capture small, soft features such as chamfers, bevels, and fillets that are pervasive in real-world designs as well as the ability to handle arbitrarily large point clouds (and, ergo, arbitrarily fine features).

Through extensive experiments, we evaluated our approach on standard CAD benchmarks (ABC), as well as more challenging datasets (F360+, Thang3D), that contain unconventional and irregular geometries. Across all settings, our model consistently outperformed prior methods such as PCED and NerVE on edge classification tasks, achieving greater accuracy, precision, and recall, particularly for shapes with complex or soft features. Ablation studies highlighted the critical role of curvature gradients, global context, and high-quality normals, demonstrating the necessity of integrating both local geometric cues and long-range structural information. When coupled with our deterministic flood-fill clustering algorithm, our predictions provided better segmentation than common ML-based methods (e.g., ParSeNet), delivering higher mIoU and fewer incorrect segments.

These results highlight the generalizability of our method: by grounding inference in geometric and topological properties rather than in human-imposed categories, our framework extends beyond conventional benchmarks to capture the rich diversity of real-world shapes. The ability to decompose organic or unlabeled parts into meaningful geometric segments positions our approach as a foundational tool for downstream tasks in reconstruction, generative modeling, and design automation.

## 7 FUTURE WORK

Currently, our clustering algorithm is efficient and fast, but it could be improved beyond heuristic methods to be more noise resilient for even better results. Looking forward, the complex segments and outlines obtained from our primitive-less geometric segmentation hold promising potential for both reconstructing and generating a much more diverse set of 3D shapes in parametric forms than what has been previously done. It also allows more generalized processing of 3D data and can be used to potentially automatically segment larger, currently unlabeled datasets. In other applications, simply being able to encode outlines as identifiers of a shape could provide crucial information as to what different use-cases of the shape could be (as in the earlier example of automating furniture assembly or creating a shape from geometric constraints). These advances open the door to more versatile and function-aware 3D shape understanding.

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

# A APPENDIX

We show further examples of how our edge detection algorithm is capable of identifying soft edges, fine features, and spline-fit edges below.

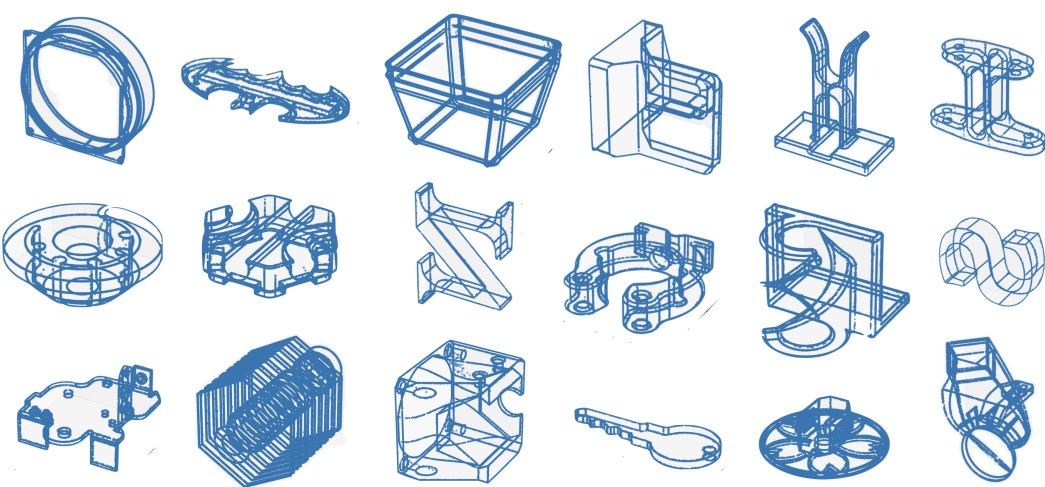

Figure 7: Examples of extracted outlines from the Thang3D dataset using our method.

Below is a detailed outline of our point-cloud segmentation algorithm.

---

**Algorithm 1** Boundary-Aware Segmentation of Point Clouds

---

1: **Input:** points $P \in \mathbb{R}^{N \times 3}$, normals $N \in \mathbb{R}^{N \times 3}$, boundary mask $B \in \{0,1\}^N$, params $(k, \tau_{\text{reject}}, r_{\text{bdry}}, \theta_0, \lambda)$
2: Normalize $N \leftarrow N/\|N\|_2$ row-wise
3: Build KD-tree on $P$ for kNN queries
4: Build KD-tree on boundary points $P_B = \{p_i : B_i = 1\}$ and compute $d_i = \text{dist}(p_i, \partial)$
5: Initialize edge set $E \leftarrow \emptyset$
6: **for** each $i \in \{1, \ldots, N\}$ **do**
7:     **for** each $j$ in kNN$(i)$ with $j > i$ **do**
8:         $m \leftarrow \frac{1}{2}(p_i + p_j)$, $d_m \leftarrow \text{dist}(m, \partial)$
9:         **if** $B_i \oplus B_j = 1$ **or** $\min(d_i, d_j) \leq \tau_{\text{reject}}$ **or** $d_m \leq \tau_{\text{reject}}$ **then**
10:             **continue**
11:         **end if**
12:         $w \leftarrow \|p_j - p_i\|_2$
13:         **if** $\min(d_i, d_j, d_m) \leq r_{\text{bdry}}$ **then**
14:             $\theta \leftarrow \arccos(\text{clip}(n_i^\top n_j, -1, 1))$
15:             **if** $\theta > \theta_0$ **then**
16:                 $w \leftarrow w \cdot \left(1 + \lambda \frac{\theta - \theta_0}{\pi - \theta_0}\right)$
17:             **end if**
18:             $w \leftarrow w \cdot \max\left(1, \frac{r_{\text{bdry}}}{\max(10^{-12}, \min(d_i, d_j, d_m))}\right)$
19:         **end if**
20:         Add edge $(i, j, w)$ to $E$
21:     **end for**
22: **end for**
23: Symmetrize edges: $E \leftarrow E \cup \{(j, i, w) : (i, j, w) \in E\}$
24: Build adjacency $A$ from $E$ (binary or weighted)
25: Compute connected components of $A$, yielding labels $\ell \in \{1, \ldots, C\}^N$
26: **Return:** labels $\ell$, edge list $E$

---

Our results presented in the paper for accuracy, F1, precision, and recall for the NerVE model assumed that predicted edge points within 0.05 of the true points were 'correct.' However, as the threshold is tightened, performance vastly drops (F1, precision, and recall being computed over the entire set of points). Accuracy becomes 'better' when the threshold gets lower because as the band for boundary points tightens, fewer non-boundary points count as false positives within the band. These results highlight that in reality, the average distance between the predicted edges and the actual edges is roughly **0.1**, which is a fairly large margin of error (5%) since the shape is scaled to be within the unit sphere.

| NerVE Threshold | Acc. ↑ | F1 ↑ | P ↑ | R ↑ |
|---|---|---|---|---|
| 0.1 | 0.52 | 0.65 | 0.71 | 0.59 |
| 0.05 | 0.76 | 0.34 | 0.41 | 0.25 |
| 0.01 | 0.93 | 0.09 | 0.04 | 0.03 |
| 0.005 | 0.94 | 0.03 | 0.01 | 0.009 |
| 0.001 | 0.95 | 0.0003 | 0.0002 | 0.0005 |

Table 6: Table showing NerVE's performance degradation as the threshold for what is considered a 'correct' edge is tightened. Evaluated on the F360+ dataset.

