# OpenReview forum: "Generalizable 3D Edge Detection For Soft & Hard Features"
_ICLR.cc/2026/Conference — Submitted to ICLR 2026_

### Official Review · Reviewer_CNhq · 2025-10-22

**Soundness:** 3
**Presentation:** 3
**Contribution:** 2
**Rating:** 4
**Confidence:** 3

**Summary:**

This paper proposes a 3D point cloud edge detection algorithm that can effectively detect the geometric components of complex 3D objects without the need for raw annotations or semantic labels. The method achieves state-of-the-art (SOTA) performance on multiple datasets, which demonstrates its effectiveness.

**Strengths:**

1.This paper improves edge detection performance by introducing abundant geometric features. Even though the proposed architecture is simple, it can still achieve optimal results on multiple datasets.

2. The formulas in the paper are complete, and the visualizations are clear.

**Weaknesses:**

1.There is a lack of reports on running time and computational overhead.

2.The ablation experiments lack the part related to the skip branch, making its role unknown.

3.The paper has poor readability and a rather confusing logic.

4.Using more explicit geometric information as input has been adopted by many previous methods and proven effective; therefore, the innovation of this method is relatively weak.

**Questions:**

1.This paper mentions dividing each point cloud into non-overlapping subsets of 10k points. Has this been proven to be a favorable choice through experiments? Will different point quantities affect the model performance?

---

> ### Author Response · Authors · 2025-11-21
> **This response contains new requested ablations for skip-branch and chunking, also we clarify our contribution and lack of existing work in this particular task.**
>
> As reported to reviewer x1iF, pre-processing time tends to scale linearly with more points as the pre-processing is all KNN-based, it takes roughly 30 seconds to pre-process a sample of 1.3M points on a L40S GPU. Training on an H100 averaged about 1.87 seconds per batch with a batch size of 16 samples, averaging roughly 340k points per sample. Inference time is roughly 1.1 seconds per batch of 16 with an average point size of nearly the same. We will add the run time values in the full paper.
> ***
> Skip-branch ablation results (will be added to the paper):
>
> Epochs to Convergence: 90
>
> Acc = 0.99
>
> F1 = 0.82
>
> Precision = 0.83
>
> Recall = 0.83
> ***
>
> Convergence is defined as no increase in the accuracy or F1 validation set scores after 10 epochs (usually we start to see those validation statistics start to decrease roughly 2-3 epochs after they hit their maximum). As noted in the evaluation section, the chunks do not overlap.
> ***
> Chunk Ablation Results (will also be added to paper):
>
> 8k Chunks
>
> Acc=0.98
>
> F1=0.84
>
> Precision=0.81
>
> Recall=0.89
>
> Epochs to convergence: 24
>
> ***
> 12k Chunks
>
> Acc=0.98
>
> F1=0.78
>
> Precision=0.75
>
> Recall=0.83
>
> Epochs to convergence: 26
> ***
>
> We would greatly appreciate it if the reviewer could point out the specific portions where the readability could be improved. We are more than happy to revise those sections. As mentioned in previous responses, we will also be adding additional information throughout the paper, which we hope will help clarify the presentation.
>
> We have not identified prior work that uses explicit geometric information (such as local normals and curvature) within an architecture designed to perform precise edge detection across a broad and diverse range of shapes. We are very eager to compare our method with any competing approaches that can operate on these datasets, and we have included comparisons with every method for which code is publicly available or provided to us that is compatible with our kind of point clouds. Unfortunately, as noted in our responses to reviewers x1iF and Q4Cf, the authors of some relevant methods were unable or unwilling to share their code, limiting our ability to evaluate them. We will tactfully describe the limitation in comparisons in the paper.
>
> As discussed in our responses to reviewers x1iF and Q4Cf, a key contribution of our work is the ability to recursively perform edge classification on large, dense point clouds. We will make this clearer in the paper. Our focus is on reconstructing fine features—a largely under-explored area compared to coarse or primitive-driven reconstruction. Our approach relies on local geometric information rather than primitive identification, which typically requires the entire shape to be visible to the model for accurate inference. This reliance on local features makes the recursive structure possible. By contrast, methods that group points into primitives, such as those highlighted by reviewer GtZ6, would face significantly greater difficulty in performing this task on large, primitive-less point clouds.

---

### Official Review · Reviewer_GtZ6 · 2025-10-31

**Soundness:** 2
**Presentation:** 3
**Contribution:** 2
**Rating:** 2
**Confidence:** 5

**Summary:**

The paper addresses the problem of 3D edge detection in point cloud representations of shapes, emphasizing how identifying regions of high or moderate curvature, such as ridges, valleys, or protrusions, can enhance the accuracy of downstream geometric or primitive segmentation.

The authors propose an edge detection framework that leverages low-level geometric cues, including point coordinates, normals, mean curvature, and curvature gradients at multiple scales. These features are processed by a point-based neural network that predicts, for each point, the probability of belonging to an edge/boundary region. To efficiently handle large point sets (hundreds of thousands of points), the model operates on 10K-point patches while maintaining a learned global context vector that is iteratively updated across patches to preserve global geometric consistency.

After per-point boundary prediction, the method applies a flood-filling clustering algorithm to segment the shape into distinct geometric components based on boundary connectivity. The model is trained using BREP-based CAD datasets, with dense point sampling around high-curvature regions to better capture fine geometric details.

The proposed approach is evaluated on edge detection and geometric segmentation tasks, where it outperforms existing methods.

**Strengths:**

The paper addresses a challenging problem in point cloud analysis, performing primitive-free geometric segmentation without relying on predefined semantic/primitive labels. By framing the task as boundary region detection, the approach eliminates the need for labeled part data beyond edge annotations, thereby improving generalizability across datasets and shape types. The use of low-level geometric features such as curvature and curvature gradients enables the network to accurately capture regions of high convexity or concavity, including soft transitions like fillets and chamfers. Furthermore, the introduction of a global context vector that is recurrently updated across point patches helps maintain global geometric consistency, allowing the model to process large point clouds while preserving overall shape coherence.

**Weaknesses:**

While the paper presents a practically effective method, its architectural and conceptual novelty is limited. The patch-based point cloud processing strategy closely resembles EC-Net, which also processes shapes in local patches, while the edge detection formulation is conceptually similar to PB-DGCNN [1].

From an evaluation perspective, several aspects raise concerns about fairness and completeness. The method is compared primarily against NerVE and PCED, yet these baselines are designed for much smaller point sets (around 20K points), whereas the proposed model is trained and tested on significantly denser clouds (approx. 200K points). This discrepancy potentially biases the results, since higher sampling density naturally enables better capture of fine geometric details. Moreover, although the paper states (lines 337–338) that experiments were conducted on the full F360+ and Thang3D datasets, it later clarifies (lines 356–358) that only about half of F360+ and one-fourth of Thang3D were actually processed. Additionally, qualitative visualizations comparing failure cases of competing methods are missing, which would have provided clearer insight into where and how the proposed approach improves over baselines.

For the segmentation benchmark, the evaluation includes only ParSeNet, while other relevant and publicly available alternatives such as HPNet [2], PrimitiveNet [4], and SED-Net [3] are omitted. The same issue regarding point cloud resolution mismatch arises here, as ParSeNet was trained on lower-resolution inputs (10K points). Furthermore, in the proposed flood-filling clustering stage, it remains ambiguous whether the adjacency matrix is treated as binary or weighted, and how this choice affects the resulting segmentation quality. Finally, additional quantitative and qualitative comparisons would strengthen the experimental section and better substantiate the claimed superiority of the proposed method.

Finally, the paper claims that a CNN model is applied, but essentially an MLP PointNet-based encoder is utilized.


[1] Loizou M. et al., "Learning part boundaries from 3D point clouds." Computer Graphics Forum. Vol. 39. No. 5. 2020.

[2] Yan S. et al., "Hpnet: Deep primitive segmentation using hybrid representations." Proceedings of the IEEE/CVF international conference on computer vision. 2021.

[3] Li Y. et al., "Surface and edge detection for primitive fitting of point clouds." ACM SIGGRAPH 2023 conference proceedings. 2023.

[4] Huang J. et al, "Primitivenet: Primitive instance segmentation with local primitive embedding under adversarial metric." Proceedings of the IEEE/CVF International Conference on Computer Vision. 2021.

**Questions:**

- How do the different neighborhood scales used for computing curvature and curvature gradients affect the model’s performance in edge detection??
- What are the values of the hyperparameters $k$, $\tau_{\text{reject}}$, $r_{\text{bdry}}$, $\theta_0$, and $\lambda$ in the boundary-aware segmentation step, and how do variations in these parameters influence the final segmentation results?

---

> ### Author Response · Authors · 2025-11-21
> **Prior primitive/semantic-based methods for segmentation are challenging to compare to, and often impossible to do with our data (though prior work indicates full-reconstruction does not perform well).**
>
> Thank you for this in-depth review!
>
> You are absolutely right that many existing methods operate on significantly smaller point sets, and one of their known limitations is that they often do not scale well. As we noted in our responses to reviewers x1iF and Q4Cf, an important aspect of our contribution is enabling recursive edge classification on large point clouds. This is feasible because our approach relies on local geometric cues rather than primitive identification, which typically requires the full shape to fit within the model for accurate inference. Our feature design makes this recursive structure possible.
>
> We also want to clarify our use of ParSeNet. For fairness, we allowed it to classify points iteratively by repeatedly feeding in uniformly distributed samples of the full point cloud until all points were covered. The reported results come from this iterative setup, and we can make this clearer in the paper. Adjusting ParSeNet to directly accept much larger point sets would require substantial architectural changes. In contrast, a key strength of our method is that it can naturally handle larger point sets than prior work.
> For HPNet, PrimitiveNet, and SED-Net, these methods are all designed around primitive-based segmentation and reconstruction. The core challenge we address, however, is handling shapes whose segments do not conform well to primitives—this is true for much of the data in both of our evaluation datasets. Even if segmentation were accurate, their primitive-driven classification outputs would not align with these shapes. Moreover, these methods perform segmentation not edge detection. Our goal is specifically to identify boundary edges so we can reconstruct clean parametric CAD edges. We will make this distinction more explicit in the paper.
>
> For a direct comparison, we would first need to adapt these models to process large point sets simultaneously (which their primitive-based designs typically require). This would be difficult in practice, both due to hardware constraints for point clouds that large and because these models depend on semantic or type-labeled input features (HPNet, SED-Net) that are not available for our data, or KNN-based estimators (PrimitiveNet) that assume all points are processed at once. We would then need to redesign their output layers to detect edges instead of segments—or attempt to extract edges indirectly from segment boundaries. These steps would amount to building substantially new architectures rather than evaluating the existing ones, which were designed for a different task and a narrower problem setting.
>
> We appreciate the opportunity to clarify this and will incorporate a concise version of this explanation in the revised manuscript.
> Additionally, these datasets were evaluated against Point2CAD (Liu et Al) and Point2Cyl (Uy et Al)  in Kimmel et. al for full reconstruction, both of which used HPNet for initial segmentation, the outputs performed far worse on these datasets than ABC. The numbers show only 18.9% of F360+ shapes were reconstructed correctly and 12.3% of Thang3D shapes.
>
> As stated in the evaluation section, NeRVE was tested on the full F360+ and Thang3D datasets. For PCEDNet, we were only provided with a compiled graphical-interface binary, which required manual file input. Unfortunately, the compiled version frequently crashed and these crashes tended to occur more often on more complex shapes, particularly those with spline-fit edges. We could have considered these as ‘full failure’ (accuracy, F1, precision, recall all 0), but we did not.
>
> Yes, our method is closely related to PointNet—we build directly on that architecture. Since the original PointNet paper refers to itself in its title as “PointNet: A 3D Convolutional Neural Network for real-time object class recognition,” we followed that convention even though the structure differs from a traditional CNN. We will update the paper to describe the model more clearly as an MLP PointNet-based encoder, and we will also reference PointNet’s own terminology to provide context for this choice.
>
> We train using a batch size of 16 and the Adam optimizer initialized with a learning rate of 1e-3, training until convergence as described in our response to reviewer CNhq. The hyperparameters for the segmentation component are tied to the spacing of points sampled from the surface, where we target an average nearest-neighbor spacing of 0.003. This provides us with a consistent metric for curvature-calculation neighborhoods as well, ensuring that smaller features are not overshadowed—hence our use of curvature values computed over 5, 10, and 20 nearest neighbors. This spacing also serves as the threshold for rejection. Other parameters, such as minimum and maximum edge distance, are computed automatically on a per-point-cloud basis as described in the deterministic clustering section.
>
> We will include additional images of edge-detection and segmentation failure cases in the appendix.

---

### Official Review · Reviewer_Q4Cf · 2025-11-02

**Soundness:** 2
**Presentation:** 2
**Contribution:** 2
**Rating:** 4
**Confidence:** 3

**Summary:**

This paper proposes a primitive/semantic label-independent 3D edge detection method. The main idea is to incorporate various geometric features (such as mean curvature and curvature gradient) as inputs to a PointNet-like framework to classify edge points. The method is evaluated on three datasets, demonstrating superior performance compared to selected baseline methods.

**Strengths:**

- The  proposed method achieved sota in three datasets.
- The proposed method works better on soft edges.

**Weaknesses:**

- The significance and challenges of the task is not clearly convinced.
- The baseline methods are outdated and the evaluation is insufficient. For example, the work lacks comparisons with more recent approaches [1]. Moreover, there are no comparisons with recent point cloud segmentation methods.

* [1]Liu et.al., ToG’25, HoLa: B-Rep Generation using a Holistic Latent Representation,

**Questions:**

- There is no citation for the Thang3D dataset. Is it newly collected in this work? If so, please provide more information about the dataset (e.g., number of models, annotations of ground truth).
- The authors claim that the proposed method can “provide a robust and generalizable … beyond humans.” Are there any quantitative comparisons or qualitative examples to support this claim?
- Was the proposed method also only trained on the F360+ dataset?
- What is the loss function of the proposed method? How are the segmentation results derived in Figure 6?
- What does “precise geometric components” mean? Is there any visualization?

---

> ### Author Response · Authors · 2025-11-21
> **Reconstruction of small, soft, unlabeled features is a necessary but understudied subset of parametric 3D reconstruction and generation.**
>
> Thank you for the thoughtful response!
>
> As discussed in our response to x1iF, a major contribution of our work is enabling recursive edge classification on large, dense point clouds. This is feasible because we rely on local geometric information rather than primitive identification, which typically requires the entire shape to fit inside the model for accuracy. Our focus is on reconstructing fine, soft features at high fidelity—an area that remains understudied compared to coarse/primitive-driven reconstruction.
>
> We appreciate the chance to elaborate further on the use cases, and will expand this discussion in the intro. Our approach is particularly helpful for scanned-object reconstruction, but is also valuable for converting meshes into parametric representations. For instance, though modern text-to-mesh or text-to-NeRF systems produce impressive results, their outputs are often only “mostly” correct and difficult to edit beyond iterative text prompting. By enabling conversion to editable CAD, a user could directly refine features. Existing mesh-to-CAD methods, such as DeepCAD, typically handle only simple geometric models; our method supports a much more diverse range of shapes for downstream construction, as described below.
>
> Our method also provides much richer input featurization for generative models. Current generative CAD approaches generally cannot produce fine, soft features because they are limited by the number of points they can process and primitives they can map to. By iteratively handling large point sets and capturing subtle, primitive-less geometric details, our method enables these learned features to be incorporated into downstream generative pipelines.
>
> Additional use cases are provided in our response to reviewer x1iF.
>
> Thang3D is from the same paper as the F360+ dataset by Kimmel et al., as noted in lines 196–197. We recognize that the comma placement there may cause confusion and will clarify this. Additional dataset information can be found in that paper. As described to reviewer x1iF and in our evaluation section, our method was trained solely on F360+, a CAD dataset of manufactured parts. We then evaluated it on Thang3D, which contains not only standard CAD models but also domain-diverse objects, shown in Figure 7 in the appendix. Our method performs well across this highly diverse set without any retraining or fine-tuning, and we will clarify this in the paper.
> To our knowledge, no existing work has attempted precise geometric segmentation or edge detection on organic shapes with soft features. This capability is crucial for accurate reconstruction of objects in editable parametric form.
>
> Humans can segment shapes without assigning absolute semantic labels—segments can simply reflect geometric distinctions. In Figure 1, geometric segmentation extends beyond semantic categories (e.g., “teapot handle”) or primitive forms (“sphere,” “cone”) to reflect differences in surface geometry. Similarly, in Figure 6, our method separates geometric components even when they lack a clear semantic or primitive-based label, such as the spherical corner patches of a rounded cube. These components do not correspond to any standard primitive, and a human would likely describe them using surface characteristics (e.g., “this segment is roughly 40% of a sphere’s surface”). Our model operates in the same spirit.
>
> HoLA primarily studied unconditional generation, and the authors declined to provide their code along with specific hyperparameters. For the conditioned generation component, they mention using “a PointNet-based” architecture but do not specify the exact layers or parameterization used for the conditioning vectors.
> For baseline comparisons, our datasets do not include primitive labels, and many shapes do not fit within any primitive-based system (e.g., SAMPart3D, Point2CAD). In practice, this means that preprocessing for these models often fails outright; and even when it succeeds, substantial architectural modifications would be required to support primitive-agnostic labeling as described to reviewer GtZ6. We will add this clarification to the paper.
>
> Loss is a weighted binary cross-entropy loss. We will add a statement that explicitly identifies this. Segmentation results were obtained through the process described in the Deterministic Flood-Fill algorithm part under section 4, with pseudocode in the appendix.
>
> “Precise Geometric Components” refers to components that are very small relative to the entire object (less than 2% of the largest dimension). In Figure 6, the chamfered cube edges (small rounded regions) illustrate that even small segments are correctly identified and preserved rather than merged into adjacent faces. Figures 5 and 7 show many additional examples, including subtle features such as the curvature beneath the serif of the “W” object (Figure 5) and the threading on a screw mount (Figure 7). We can include even more visualizations in the appendix if helpful.

---

### Official Review · Reviewer_x1iF · 2025-11-03

**Soundness:** 3
**Presentation:** 3
**Contribution:** 3
**Rating:** 6
**Confidence:** 3

**Summary:**

This paper presents a primitive-less 3D edge detection method for point clouds that handles both soft features (fillets, chamfers, bevels) and hard features (sharp edges). The approach processes large point clouds (>200k points) by chunking them into 10k-point batches and maintaining a global context vector across chunks. The method uses a two-branch CNN architecture that processes geometric features including point coordinates, normals, curvature (H), and curvature gradients (∇H). A deterministic flood-fill clustering algorithm then segments the point cloud based on detected edges. The authors evaluate on ABC, Fusion360+, and Thang3D datasets, demonstrating state-of-the-art edge detection accuracy and improved segmentation performance compared to state-of-art approaches. The key contribution is achieving robust edge detection on diverse geometries without relying on semantic labels or geometric primitives.

**Strengths:**

**Originality**: The paper addresses an important gap in 3D edge detection—handling soft features that most existing methods fail on. The combination of curvature-based features with a chunked processing approach using global context is a reasonable design choice for handling arbitrarily large point clouds.

**Evaluation**: The experimental evaluation is comprehensive, including three diverse datasets (ABC, Fusion360+, Thang3D), ablation studies examining each component's contribution, and comparisons against recent baselines (PCEDNet, NerVE). The generalization capability is impressive, as demonstrated by training only on F360+ and testing on ABC and Thang3D without fine-tuning.

**Clarity**: The paper is clearly written, with most algorithmic details clearly outlined.

**Significance**: The work has practical value for CAD, manufacturing, and 3D reconstruction applications where capturing fine geometric details is necessary.

**Weaknesses:**

**Limited architectural novelty**: the architecture is relatively simple making the paper read more as an engineering contribution. The reliance on hand-crafted geometric features (curvature, gradients) reduces the learning burden but also limits the model's ability to discover novel feature representations.

**Dependency on feature quality**: the ablation study shows dependence on high-quality normals (Table 4: approximated normals drops F1 from 0.87 to 0.73). For real-world scanning scenarios with noise and occlusions, normal estimation quality may be unreliable. The noise experiment shows only modest degradation at 0.002 noise level, but more extensive noise analysis across different noise types and levels would strengthen the claims. Alternatively, experiments showing results on real or simulated scanned objects could be helpful for this.

**Incomplete description and motivation for the chunking approach**: the method heavily relies on chunking the point cloud for processing, but it does not provide the motivation for why such approach is reasonable / optimal, not describe algorithmic details of the approach (how are chunks determined? is the chunk size dependent on sampling resolution? is there overlap between chunks? ablation for the chunk size? etc.).

**Incomplete experiment description and analysis**:
1. Some important details are missing: dataset splits, training hyperparameters beyond epochs, convergence criteria, heuristic algorithm hyperparameter selection.
2. Why is it important to have the "skip branch" in the proposed architecture? Ablation study is necessary.
3. The segmentation comparison is limited to only ParSeNet—more recent methods like SpelsNet are mentioned but not compared.
4. Computational cost analysis is missing—how does processing time scale with point cloud size? What are training / inference times?

**Questions:**

Some of my questions and suggestions are outlined above, in the Weaknesses section.

Additional questions are below.
1. Feature engineering vs. learning: how much of the performance gain comes from the curvature features versus the neural architecture? Have you tried end-to-end learning without pre-computed curvature features using a larger network? Similarly, the approximated normals ablation shows significant degradation. Have you experimented with learning-based normal estimation methods?
2. Computational efficiency: what are the runtime comparisons against PCEDNet and NerVE?
3. Failure cases: are there specific geometric configurations where your method fails?
4. Clustering hyperparameters: how were the flood-fill algorithm parameters selected? Are they dataset-specific or can one set of hyperparameters work across all datasets?

---

> ### Author Response · Authors · 2025-11-21
> **We'll add more details related to training, also the use of deterministic features makes our approach more generalizable.**
>
> Thanks for your clear and insightful review!
>
> We’ll add full training details, parameter-selection process, and hardware information to the paper. We’ll also include a clearer description of the chunking procedure. Pre-processing time tends to scale linearly with points as pre-processing is KNN-based, it takes roughly 30 seconds to pre-process a sample of 1.3M points on a L40S GPU. Training on an H100 averaged about 1.87 seconds per batch with a batch size of 16 samples, averaging roughly 340k points per sample. Inference time is roughly 1.1 seconds per batch of 16 with similar points per sample.
>
> Skip branch and chunking variation ablation results (will be added to paper) are shown in response to CNhq.
>
> Earlier experiments were initially conducted without any curvature input – we gradually increased both the network depth and layer count. These prior, curvature-less versions performed well on “hard features” (features with sharp normal changes) but were unable to capture the soft features we aimed to reconstruct. A key contribution of our work is the ability to handle much larger point clouds by combining an architecture designed for local information with feature inputs that also rely on locality. In contrast, primitive-labeling-based approaches require the model to view at least an entire feature—if not the whole shape—at once, which is computationally prohibitive at scale. The emphasis on local geometric cues is what makes our architecture feasible.
>
> While ML-discovered features can be powerful, it also intrinsically biases networks to only perform well on data that has similar features to training data. This limitation is visible in the performance drop of PCED and NeRVE when evaluated on F360+ and Thang3D (Table 3), as well as the inability of Point2CAD and Point2Cyl to reconstruct shapes in those datasets in the ICML evaluation by Kimmel et al. The same pattern appears in ParSeNet: when applied to the F360+ dataset—which is still within the CAD domain—its segmentation results (Table 5) are significantly lower than its results on the DeepCAD dataset used in its original paper.
> Curvature calculations and curvature-based features, however, are universal. As described in the evaluation section, our method was trained only on F360+, a CAD dataset of manufactured parts. Thang3D includes typical CAD objects but also organic and varying domain shapes—bananas, cats, letters, flowers, Batman symbols—as shown in Figure 7. Our method continues to perform well on this diverse dataset without any retraining or fine-tuning. This robustness is likely a direct result of avoiding dataset-specific learned features. We will highlight this more clearly in the paper. In our method, failure modes still tend to be shapes with many organic, spline-fit lines that all meet at a specific point (i.e. ‘corners’ that are not right-angle corners).
>
> SpelsNet unfortunately did not release code nor the hyperparameters required to accurately recreate their model, we did reach out to the authors.
>
> We are mainly interested in reconstructing precisely scanned shapes and/or extracting small features for shape generation later on. We found 0.002 is the average scanning noise deviation for normal approximation for a ‘high-end hobbyist’ scanner per (per Franaszek et al, as cited in the paper). Real industry scanners are much more precise. Additionally, we will make this clear in our paper but our points are initially sampled from a mesh representation of a BREP object (i.e. a triangulated object). While this intrinsically introduces noise from the triangle approximations, it presents perhaps one of the greatest use-cases for our method – converting a mesh object into an editable, parametrized CAD object reliably with small features preserved. In this case, you can sample an infinite amount of points from the surface (as we did) with fairly accurate normals. All of our normals are read from the approximated triangulated meshes, with Gaussian noise applied as an additional component. Unfortunately, scanning real objects is a bit of an expensive experiment to run – we’d love to try this in the future though! We can clarify this in the paper.
> Most ML-based normal estimation is worse than scanned normal estimation:
>
> Nesti-Net (CVPR’19) on the PCPNet benchmark (unoriented normals):
> overall average 12.41°; by corruption level: clean 7.06°, low noise 10.24°, medium 17.77°, high 22.31°; density stripe 8.64°, density gradient 8.95°.
>
> AdaFit (ICCV’21): clean 5.19°, low 9.05°, med 16.45°, high 21.94°, stripe 6.01°, gradient 5.90°, avg 10.76°.
>
> HSurf-Net (NeurIPS’22): clean 4.17°, low 8.78°, med 16.25°, high 21.61°, stripe 4.98°, gradient 4.86°, avg 10.11°.
>
> When it comes to normal estimation from point clouds, we want to do it only relative to the neighbors it can see, we could definitely experiment with more robust ways of fitting normals though in future work.
>
> Clustering hyperparameters are discussed in response to reviewer GtZ6.

---

### Author Response · Authors · 2025-11-30
**Summary of Reviews and Rebuttals for AC**

We thank the AC for taking the time to review our paper given these extenuating circumstances.   Below is a summary of key points from our rebuttals.

---

## **Strengths (from our reviewers)**

1. **Addresses a major gap in 3D edge detection:**
   Focuses on soft geometric features (e.g., fillets, chamfers) that most existing detectors struggle with. (x1iF, GtZ6)

2. **Generalizable, primitive-free segmentation:**
   Reframes the task as boundary-region detection instead of relying on predefined primitives or semantic labels, requiring only edge annotations and improving cross-dataset generalizability. (GtZ6, x1iF)

3. **Rich geometric cues:**
   Uses curvature, curvature gradients, and other low-level features to capture both sharp and soft transitions. Highly generalizable. (GtZ6, CNhq)

4. **State-of-the-art performance:**
   Achieves SOTA results on three diverse datasets. (x1iF, Q4Cf)

5. **Strong performance on soft features:**
   Shows substantial improvements where prior methods typically fail. (CNhq, x1iF, Q4Cf)

6. **Simple yet effective architecture:**
   Performs competitively due to abundant geometric features and coherent global processing. (CNhq, GtZ6)

7. **Clear presentation:**
   Provides complete mathematical formulations, detailed algorithms, and clear visualizations supporting reproducibility. (CNhq, x1iF)

---

## **Concerns Addressed (to be added/clarified in the paper)**

 **1. Novelty and contribution**
- No prior method we have seen identifies **soft features** and edges in point clouds or handles **arbitrarily large point sets**.
- Primitive-less feature detection in prior work is rare and limited to **hard** edges.
- By relying on geometric properties rather than primitives, our method able to iteratively classify many more points than prior approaches.  This architecture would be impossible if we used primitives.
  *(See response to GtZ6.)*

 **2. Range of use cases**
- Supports scanned-object reconstruction, edge detection, and mesh-to-CAD conversion.
- Modern text-to-mesh/NeRF systems produce good but difficult-to-edit shapes; converting them to CAD enables direct refinement.
- Existing mesh-to-CAD models (e.g., DeepCAD) only handle simple shapes, while ours generalizes to complex geometry (e.g., Thang3D).
- Also enriches feature inputs for generative CAD models, which currently struggle with fine or soft features due to primitive and point-count limitations.
  *(See response to Q4Cf.)*

**3. Generalizability through deterministic geometry calculations.**
- Using deterministic curvature and gradient extraction yields far more generalizable edge detection, ML-learned features bias the models to only perform well on datasets with similar features.
- Metrics remain strong on unseen datasets even when training only on Fusion360+, while existing methods degrade (see Table 3 in paper).
  *(See response to x1iF.)*

**4. Additional ablations**
- Added ablations justifying skip-branch design and chunk size choices.  *(See response to CNhq.)*

**5. Expanded training details**
- Added more detailed hardware and timing information.  *(See response to x1iF.)*

**6. Why comparison to primitive-based methods is intrinsically flawed**
- HPNet, PrimitiveNet, and SED-Net are not designed for large pointclouds; this is a weakness of primitive-based approaches and would require heavy re-engineering of input layers to do true comparison.
- Our shapes were selected specifically because they **do not** fit common geometric or semantic primitives used by prior methods.
- Primitive-based methods perform segmentation - not boundary detection - requiring heavy custom post-processing to compare edges.
- Such modifications would constitute building **new models**, not comparing existing ones.
  *(See response to GtZ6.)*

---

Thank you again, and best of luck in this challenging situation.

---

### Meta-Review · Area_Chair_eNW6 · 2026-01-07

**Summary:**

The reviewers raised concerns about the limited novelty and insufficient experiments. Both main concerns remain unresolved after the rebuttal. Therefore, I recommend rejecting this paper.

**Reviewer Concerns:**

- Reviewer x1iF: Limited architectural novelty, Incomplete experiment description and analysis
- Reviewer Q4Cf: Unconvinced of significance, insufficient evaluation,  lacks comparisons with more recent approaches
- Reviewer GtZ6: limited novelty, unfair comparison, limited comparisons
- Reviewer CNhq: missing analysis of model complexity, insufficient ablation experiments, limited novelty

**Reviewer Scores:**

- Reviewer x1iF: No
- Reviewer Q4Cf: No
- Reviewer GtZ6: No
- Reviewer CNhq: No

---

### Decision · Program_Chairs · 2026-01-26

Reject